# Comparison of Indexes to Measure Comorbidity Burden and Predict All-Cause Mortality in Rheumatoid Arthritis

**DOI:** 10.3390/jcm10225460

**Published:** 2021-11-22

**Authors:** Yun-Ju Huang, Jung-Sheng Chen, Shue-Fen Luo, Chang-Fu Kuo

**Affiliations:** Division of Rheumatology, Allergy and Immunology, Chang Gung Memorial Hospital, Taoyuan 333, Taiwan; b9502071@cgmh.org.tw (Y.-J.H.); rschen0404@gmail.com (J.-S.C.); lsf00076@gmail.com (S.-F.L.)

**Keywords:** rheumatoid arthritis, Charlson Comorbidity Index, Elixhauser Comorbidity Index, Multimorbidity Index, Rheumatic Disease Comorbidity Index, comorbidity, mortality, comorbidity index, comorbidity burden

## Abstract

Objectives: To examine the comorbidity burden in patients with rheumatoid arthritis (RA) patients using a nationwide population-based cohort by assessing the Charlson Comorbidity Index (CCI), Elixhauser Comorbidity Index (ECI), Multimorbidity Index (MMI), and Rheumatic Disease Comorbidity Index (RDCI) scores and to investigate their predictive ability for all-cause mortality. Methods: We identified 24,767 RA patients diagnosed from 1998 to 2008 in Taiwan and followed up until 31 December 2013. The incidence of comorbidities was estimated in three periods (before, during, and after the diagnostic period). The incidence rate ratios were calculated by comparing during vs. before and after vs. before the diagnostic period. One- and 5-year mortality rates were calculated and discriminated by low and high-score groups and modified models for each index. Results: The mean score at diagnosis was 0.8 in CCI, 2.8 in ECI, 0.7 in MMI, and 1.3 in RDCI, and annual percentage changes are 11.0%, 11.3%, 9.7%, and 6.8%, respectively. The incidence of any increase in the comorbidity index was significantly higher in the periods of “during” and “after” the RA diagnosis (incidence rate ratios for different indexes: 1.33–2.77). The mortality rate significantly differed between the high and low-score groups measured by each index (adjusted hazard ratios: 2.5–4.3 for different indexes). CCI was slightly better in the prediction of 1- and 5-year mortality rates. Conclusions: Comorbidities are common before and after RA diagnosis, and the rate of accumulation accelerates after RA diagnosis. All four comorbidity indexes are useful to measure the temporal changes and to predict mortality.

## 1. Introduction

Comorbidities are conditions that coexist with a specific disease. The importance of determining comorbidity in the routine care of rheumatoid arthritis has been recognized because it may aid in predicting disability, medical cost, survival time, and mortality [1,2,3,4]. Systematic quantification of the comorbidity burden is essential for clinical management of index diseases. Currently, there are several comorbidity indexes that have been developed for various purposes. The simple method is to use the sum of each comorbidity score, whereas the more complicated way is to aggregate and weight specific comorbidities to measure the burden and impact of them on the index disease management [5,6,7,8]. The selection of different comorbidity indexes is according to the purpose of the study, type of data available (administrative data or self-report data), and outcome of interest. These tools were decided for research studies, they are not embedded widely in electronic health record systems, and there is no consensus or guidelines about how these should be used in clinical practice at this point.

The Charlson Comorbidity Index (CCI) and the Elixhauser Comorbidity Index (ECI) are general-purpose comorbidity indexes while the Multimorbidity Index (MMI) and the Rheumatic Disease Comorbidity Index (RDCI) are designed specifically for patients with rheumatic diseases. They are similar in core comorbidity categories but are validated against different patient outcomes. For example, CCI was validated against mortality, hospital stay, functional disability, and healthcare utilization [9,10,11,12]. ECI uses 30 comorbidities (17 comorbidities from CCI) to predict hospital stay, cost, and mortality. A further revised weighted version was developed, which predicted mortality with better predictive ability than the original ECI [13,14,15] and CCI [16,17]. MMI was explicitly developed for RA patients based on the health-related quality of life (HRQol) [18]. The RDCI was developed with 11 comorbid conditions for patients with RA, non-inflammatory rheumatic disorders, systemic lupus erythematosus, and fibromyalgia to evaluate their quality of life using self-reported data [19] The RDCI can also be used to predict physical disability and mortality [20]. They are useful for clinicians to measure comorbidity burden and provide various predictive values to help the decision making for patient management.

There are few studies to compare comorbidity indexes in patients with RA (Appendix A). This study aims to understand the temporal changes of comorbidity indexes with the diagnosis of RA and their comparability in mortality prediction. We conducted the study using the National Health Insurance (NHI) Database, which covers the entire population since 1998; therefore, it is suitable to delineate the temporal changes of comorbidity indexes and their predictive power for mortality.

## 2. Materials and Methods

### 2.1. Study Design

This retrospective study compares the trend, measurement time, and discriminant capacity of different comorbidity indexes in RA patients. This study was approved by the Institutional Review Board at the Chang Gung Memorial Hospital, Taiwan, and was conducted in accordance with the Declaration of Helsinki. The requirement for signed informed consent was waived since the data presented in this paper were based on the NHI database, which is fully encrypted to prevent confidentiality leak. The data can only be accessed in the dedicated computer center in the NHI administration, the data holder; the study team does not hold the original data.

### 2.2. Data Sources and Patient Cohort

The primary data source came from the NHI Database, which contained registration information and original claims data of all beneficiaries of NHI in Taiwan since its establishment in 1995. The NHI is a single-payer mandatory enrolment system. Therefore, the coverage rate is virtually universal (approximately 99%). All entries for an individual are linked by a unique personal identifier assigned to each Taiwanese resident, allowing accurate linkage of records from the registration files and the original claims data. Before release for research, personal identifiers are de-identified to ensure confidentiality. The registry of beneficiaries, one of the registration files, contains details of demographics, residence, kinship relationships, occupation categories, insurance status, and insurance amount of all beneficiaries of NHI. Claims data on all outpatient visits, inpatient care, and pharmacy dispensing were recorded in specific datasets with information such as dates of events, medical diagnoses, medical expenditure, and details of prescriptions, operations, examinations, and procedures. The NHI database is linked to the National Death Registry as well, so to ascertain survival status. The availability of comprehensive medical records from national medical institutions and the focus on chronic comorbidities minimizes the risk of missing comorbidities of interest, but the accuracy of coded diagnoses can be suboptimal. Standard procedures used in administrative claims studies, such as requiring 2 codes at least 30 days apart, were used to improve reliability of the coded diagnoses.

Patients with RA were identified from the catastrophic illness registry, which records patients with specified disorders eligible for copayment waivers. A panel review of the applicant’s clinical data is mandatory for the issuance of such benefits; therefore, the case definition is stringent. Patients must fulfill the American Rheumatism Association 1987 revised classification criteria [21] to be included in the registry and receive waivers. We identified 24,767 RA patients diagnosed between 1998 and 2008 and followed them until 31 December 2013, or until patient death. The medications in Table 1 are initial medications used within three months of RA diagnosis. We used the propensity score matching to obtain general patients (99,068 non-RA patients) as the control group.

### 2.3. Comparison of the Four Comorbidity Indexes

The comorbidity was ascertained using the International Classification of Diseases, Ninth Revision, Clinical Modification (ICD-9-CM) code [22]. The exact formula and comparison of the four indexes are shown in Appendix A The overall Dartmouth-Manitoba’s CCI score was the sum of the 17 weighted comorbidity categories assigned a score of 1, 2, 3, or 6 [12]. The weighted ECI ranged from −19 to 89 and consisted of 30 comorbidities [15]. The MMI consisted of 40 comorbidities with a score range of 0–38 in the weighted version and 1–12 in the unweighted version [23]. We used the unweighted version in this study because the unweighted/counted MMI can perform as good as the weighted version [18] (Appendix A). The RDCI consisted of 11 comorbidities ranging from 0 to 9 [20]. The comparison of the four comorbidity indexes is described in Appendix A, which included the number of diseases, validated outcomes, unique and common comorbidity composites, and research about RA. The ECI consists of up to 30 comorbidities; however, the RDCI only consists of 11 comorbidities. Specific organ involvement is considered the major comorbidities in all indexes, such as lung disease, stroke, cancer, heart failure, diabetes, and coronary heart disease. It is worth noting that fracture is included in the RDCI, but renal and liver diseases are excluded. Since CCI and ECI both have a category of connective tissue disease, including RA, the score in this category was not counted in the subsequent analysis.

### 2.4. Statistical Analysis

We calculated the four comorbidity indexes at diagnosis and annual percentage change of the comorbidity index in RA patients. Using conditional Poisson regression, we subsequently estimated the IRRs and their 95% confidence intervals by comparing the IRs during the diagnostic and post-diagnostic periods with the IRs during the pre-diagnostic period in the same patients [24]. We identified patients with a comorbidity score in the quintile of highest scores of each comorbidity index assessed during RA diagnostic period (the period of 4 months before and after the initial diagnosis). This group of patients was termed “high-score group” and the cut-off value was 2 for CCI, 3 for ECI, 1 for MMI, and 2 for RDCI. The high-score group included patients in the top 20% who obtained the highest scores in the four comorbidity indexes. The 1-year and 5-year mortality rates were calculated for the low and high-score groups for each index. The Cox proportional hazards model was used to calculate hazard ratios (HR) and 95% confidence interval (CI) for death associated with a high comorbidity index. The predictive capability for the 1-year and 5-year mortality rates by the four comorbidity indexes are discriminated by using the Harrell’s c-statistics and Akaike information criteria (AIC). We also compared the performance of the four comorbidity indexes to predict 1- and 5-year mortality by the Harrel’s c-statistic (which is equivalent to area under the receiver-operator curve in the logistic regression). The impact of the comorbidity index was compared by using the weighted Kaplan-Meier survival estimates. The baseline comorbidity indexes are calculated from all comorbidities occurring during RA diagnostic period (the period of 4 months before and after the initial diagnosis). All statistical analyses were carried out by SAS 9.4 (SAS Inc., Cary, NC, USA).

## 3. Results

### 3.1. Baseline Characteristics of RA Patients

Among the 24,767 patients with RA, the mean age ± standard deviation at diagnosis was 50.2 ± 15.7 years, and the male was 20.8%. The median (interquartile range) follow-up duration was 8.6 (6.5–10.8) years. The patient characteristics and comorbidity compositions, according to different definitions of four comorbidity indexes, are shown in Table 1. The distribution of prevalent comorbidity items is different among different comorbidity indexes.

### 3.2. Four Comorbidity Index Increased after RA Was Diagnosed

The mean score at diagnosis was 0.8 in CCI, 2.8 in ECI, 0.7 in MMI, and 1.3 in RDCI. The annual percentage changes of CCI, ECI, MMI, and RDCI after the RA diagnosis were 11.0%, 11.3%, 9.7%, and 6.8%, respectively (Figure 1). The incidence rates and incidence rate ratios for the accumulation of any disease in the comorbidity index before, during, and after the diagnostic period in patients with newly diagnosed RA were demonstrated in Table 2. The incidence rate of any increase in comorbidity score during the diagnostic periods was 0.013, 0.064, 0.017, and 0.030 per patient-months for CCI, ECI, MMI, and RDCI, respectively. Compared with the before diagnostic period, the incidence rate ratio was 1.80 (1.68–1.92), 2.77 (2.67–2.87), 1.33 (1.26–1.40), and 1.47 (1.41–1.54). The patter was similar for the comparison between the ‘after’ the diagnosis periods and ‘before’ the diagnostic period.

### 3.3. Mortality Risk Associated with High Comorbidity Index

Among the 24,767 RA patients, 3531 (14.2%) died during the observation time. The mortality rate and hazard ratio in high and low-score groups are shown in Table 3 [18,20,25,26,27]. The one-year mortality rate was 2.4–3.2 per 1000 people in the low-score group and 13.1–22.8 per 1000 people in the high-score group according to four comorbidity indexes. The 5-year mortality rate was 30–41 per 1000 people in the low-score group and 117.1–175.8 per 1000 people in the high-score group according to the four comorbidity indexes. The one-year mortality risk in the high-score group was two to four times higher than that in the low-score group. (adjusted HR = 2.5–4.3) The five-year mortality risk in the high-score group was two times higher than that in the low-score group (adjusted HR = 2.5–4.3).

### 3.4. The Predictive Ability for Mortality among the Four Comorbidity Indexes

The comorbidity demonstrated a significant impact on disease-specific survival by CCI, ECI, MMI, and RDCI in the high-score group than in the low-score group, as shown in the weighted Kaplan-Meier survival curve (Appendix A). All of the log-rank test’s *p*-values were < 0.001. The discrimination for 1- and 5- year survival in the four comorbidity indexes were all acceptable (Table 4). CCI showed a slightly better predictive ability than the other three comorbidity indexes. The Harrell’s c-statistic and AIC of the 1-year mortality and 5-year mortality was highest in CCI. Appendix A shows the comparison of the ROC curve of the four comorbidity indexes in the prediction of 1-year and 5-year mortality.

### 3.5. Comparison of Comorobidity Indexes in RA Patients and Control Group

RA patients had higher score of four comorbidity indexes compared with the control group (Appendix A). Similar to RA patients, the four comorbidity indexes increased with time in the control group (Appendix A). High comorbidity index score was also associated with high mortality in control group (Appendix A). The discrimination for 1- and 5- year survival in the four comorbidity indexes were all acceptable in the control group (Appendix A, Appendix A).

## 4. Discussion

This study demonstrates that all four commonly used comorbidity indexes are useful to track patient outcomes in terms of overall disease burden and mortality, in addition to their original validated outcomes. The accumulation of comorbidity burden was higher after RA diagnosis. A higher comorbidity burden in RA patients at diagnosis is associated with increased mortality for RA patients. The short- and long-term mortality prediction performed comparably well among the four indexes, and CCI seems slightly better. This study documents the importance of comorbidity assessment, and all four commonly used indexes are all good for research outcome assessment in RA patients.

### 4.1. The Factors That Cause Comorbidity Indexes of RA Increase with Time

The comorbidities accumulated faster during and after the diagnostic periods in RA patients. The possible explanation for this finding is that a broad range of physical examination, laboratory studies, and examinations are performed before prescribing disease-modifying anti-rheumatic drugs (DMARDs) to avoid any side effects; therefore, many diseases or comorbidities are usually found during the diagnostic period. There are several reasons why comorbidities may accumulate with time. First, the inflammatory process and the side effect of a drug can cause disease-associated organ damage. Second, the increase in the risk of infection and malignancy is considered due to the disease process, disordered immune surveillance, and administration of immunosuppressant drugs [28,29]. Third, the increased risk for cardiovascular diseases and osteoporosis is due to the inflammatory process and chronic glucocorticoid use [30,31,32,33]. These indexes can only increase over time as chronic conditions do not disappear, and influence the increased comorbidity indexes after rheumatoid arthritis diagnosis. Comorbidity increased with time, and therefore a comprehensive assessment for comorbidity in RA patients since the initial diagnosis of RA is important for patient management.

### 4.2. A High Comorbidity Index Predicts a High Mortality Rate

Patients with RA who have a higher comorbidity index score early in their disease course have a significantly higher 1- and 5-year mortality risk. High burden of comorbidity related to high mortality in RA patients urges the rheumatologist to actively monitor and manage modifiable risk factors during or after RA diagnosis. In our study, the risk factors and outcomes of CVD such as hypertension, diabetes, cerebrovascular disease, and coronary heart disease are among the top six comorbidities in four comorbidity indexes (Table 1). Among the multimorbidities of RA, cardiovascular diseases contribute to the mortality of RA most [34,35]. One meta-analysis concluded that the standardized mortality rate (SMR) is 1.47 (95% CI 1.19 to 1.83) in RA compared with the general population, and no decrease was seen over time in the meta-regression [36]. Therefore, modification of cerebrovascular diseases (CVD) risk factors in RA (smoking status, blood pressure, lipid values, and diabetes mellitus status) is advised by the 2015/2016 EULAR recommendation [37].

Hypertension is the top-ranking comorbidities with a prevalence rate of 18.83% during the diagnostic period in our paper. Panolula et al. showed that hypertension is prevalent in RA and under-diagnosed (39.4%) [38]. The reason for the difference in the prevalence of hypertension between our result and Panolula et al. (18.83% vs. 70.5%) is possible under-diagnosis in early RA and the usage of antirheumatic drugs such as corticosteroids, NSAIDs, cyclosporine, and leflunomide. Otherwise, hypertension is the most important modifiable risk factor for cardiovascular disease, being more common than cigarette smoking, dyslipidemia, or diabetes [39]. We should screen and monitor the patient’s blood pressure because poorly controlled hypertension will affect the mortality of RA patients.

### 4.3. Comparison of the Four Comorbidity Indexes in RA Relevant to Mortality

There is still no consensus on the optimal comorbidity index for RA to predict mortality. Mortality is one of the validated outcomes in the first study of CCI, ECI, and RDCI [40]. MMI later proved that it can be applied to mortality in community-dwelling participants in the United States [41]. The four indexes performed similarly well in predicting mortality in RA in our study. However, other studies showed different results. In 2015, England et al. compared the performance of RDCI, CDI, FCI, Elixhauser Total Score (ETS), EPS, and simple comorbidity count (COUNT) in predicting the mortality and physical functioning in a large US cohort [20]. They concluded that ETS and RDCI are preferable comorbidity index in patients with RA as they best predicted death. A recently published paper in the UK showed that, in early RA (6591 patients), both the RDCI and CCI comorbidity indexes were associated with an increased risk of all-cause mortality [42]. RDCI predicts all-cause mortality better than CCI. Due to the comparable prediction of mortality in RA in four comorbidity indexes, for prognostication, MMI or RDCI might be easier to incorporate into the outcome research clinical process due to its simple comorbidity composition (12 comorbidities and 11 comorbidities).

### 4.4. Comorobidity Indexes in Control Group

The characteristic of aging is that it is easy to acquire many chronic diseases, which means that there is a higher chance of comorbidities. Aging can cause chronic dysregulation and even further dysfunction in multiple organ systems, so that comorbidities can be found by doctors. In elderly populations, the prevalence of patients with multimorbidities increases [43]. The comorbidity index rising with time has also been demonstrated in the general population [44]. RA patients had a higher score of four comorbidity indexes compared with the control group. This reason may be due to RA patients having a higher status of inflammation compared to the general population.

### 4.5. Limitations

Some limitations must be considered when interpreting our study findings. First, misclassification of comorbidities may have occurred. Second, we did not compare the quality of life and disability among the four comorbidity indexes. The inherent constraint in the indexes includes some scores that did not include comorbidities related to disability and high disease activity, such as osteoporosis, osteoarthritis, and fibromyalgia. All four comorbidity indexes do not consider RA disease activity. In the future, we could further compare functional status and quality of life by comorbidity indexes in patients with RA.

## 5. Conclusions

In summary, in a large, Taiwan community-based cohort, we found that people with rheumatoid arthritis (RA) had a high burden of comorbidities that accumulated with time. This was found in global high comorbidity index scores and specifically in the high incidence of hypertension, ulcer, and diabetes mellitus. In this population-based cohort study, the four comorbidity indexes are useful tools for predicting all-cause mortality among RA patients. Patients with higher comorbidity index scores thus had a higher one-year and five-year mortality rate. Screening and treating comorbidities should be valued and practiced in a rheumatologist’s daily work. Furthermore, developing adequate strategies to decrease the burden of these comorbidities among RA is suggested. Close surveillance of risk factors and aggressive management of potentially reversible risk factors are necessary to increase the survival of affected patients.

## Figures and Tables

**Figure 1 jcm-10-05460-f001:**
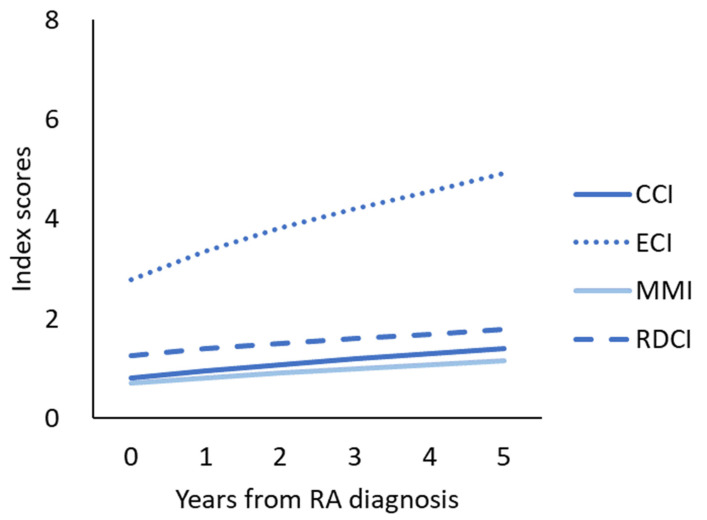
The mean scores of the comorbidity indexes according to the year of rheumatoid arthritis diagnosis. Abbreviations: Charlson Comorbidity Index (CCI), Elixhauser Comorbidity Index (ECI), Multimorbidity Index (MMI), Rheumatic Disease Comorbidity Index (RDCI), rheumatoid arthritis (RA).

**Table 1 jcm-10-05460-t001:** Characteristics of RA patients and comorbidity prevalence.

Characteristic	Value				
Male, *n* (%)	5140 (20.8)				
Age years, mean ± SD	50.2 ± 15.7				
Comorbidity indexes, mean ± SD					
CCI	0.8 ± 1.4				
ECI	2.8 ± 5.2				
MMI	0.7 ± 1.1				
RDCI	1.3 ± 1.5				
Place of residence, *n* (%)					
Urban	14,140 (57.1)				
Suburban	7333 (29.6)				
Rural	2279 (9.2)				
Unknown	1015 (4.1)				
Income levels, *n* (%)					
Quintile 1	4548 (18.4)				
Quintile 2	4038 (16.3)				
Quintile 3	6926 (28.0)				
Quintile 4	4355 (17.6)				
Quintile 5	4758 (19.2)				
Unknown	142 (0.6)				
Occupation, *n* (%)					
Dependents of the insured individuals	6561 (26.5)				
Civil servants, teachers, military personnel and veterans	1057 (4.3)				
Non-manual workers and professionals	5688 (23.0)				
Manual workers	9382 (37.9)				
Other	2079 (8.4)				
Medications					
Hydroxychloroquine	8001 (73.5)				
Azathioprine	586 (5.4)				
Methotrexate	2751 (25.3)				
Sulfasalazine	4554 (41.8)				
Leflunomide	105 (1.0)				
Cyclosporine	165 (1.5)				
TNF inhibitor	17 (0.2)				
**Comorbidity Prevalence, %**		**CCI**	**ECI**	**MMI**	**RDCI**
Ulcer or stomach problem	29.66%				V
Hypertension	18.83% 15.61%		V	V	V
Ulcer disease	15.28%	V			
Other cardiovascular	12.46%				V
Peptic Ulcer Disease excluding bleeding	11.12%		V		
Liver Disease	8.94%		V		
Lung disease	8.9%				V
Chronic Pulmonary Disease	8.18%		V		
Diabetes	7.33%			V	V
Chronic pulmonary disease	7.17%	V			
Diabetes Uncomplicated (mild to moderate)	6.27%	V	V		
Coronary heart disease	5.37%			V	
Chronic obstructive pulmonary disease	4.34%			V	
Depression	3.41% 3.22%			V	V
Asthma	3.26%			V	
Viral hepatitis	3.19%			V	
Diabetes with chronic complications	2.92%	V	V		
Cerebrovascular disease	2.74%	V			
Deficiency Anemia	2.57%		V		
Cardiac Arrhythmia	2.51%		V		
Cancer	2.45%			V	V
Any tumor	2.35%	V			
Solid Tumor without Metastasis	2.17%		V		
Congestive heart failure	2.11%	V			
Mild liver disease	1.98%	V			
Congestive Heart Failure	1.96%		V		
Valvular Disease	1.92%		V		
Stroke	1.75% 0.76%			V	V
Hypothyroidism	1.49%		V		
Renal Failure	1.36%		V		
Renal disease	1.34%	V			
Fluid and Electrolyte Disorders	1.23%		V		
Chronic Kidney Disease	1.19%			V	
Peripheral vascular disease	1.07% 0.96%	V	V		
Other Neurological Disorders	1.05%		V		
Fracture spine, hip, or leg	0.98%				V
Myocardial infarction	0.65%				V
Coagulopathy	0.63%		V		
Paralysis	0.53%		V		
Dementia	0.5%	V			
Hemiplegia	0.49%	V			
Pulmonary circulation disorders	0.43%		V		
Blood Loss Anemia	0.4%		V		
Weight Loss	0.39%		V		
Alcohol Abuse	0.36%		V		
Psychoses	0.36%		V		
Myocardial infarct	0.27%	V			
Metastatic solid tumor	0.25%	V			
Metastatic cancer	0.25%		V		
Lymphoma	0.19%		V		
Obesity	0.12%		V	V	
Moderate or severe liver disease	0.11%	V			
Drug Abuse	0.1%		V		
Diverticulitis	0.04%			V	
Acquired immune deficiency syndrome	0.02%	V	V		

Abbreviations: Charlson Comorbidity Index (CCI), Elixhauser Comorbidity Index (ECI), Multimorbidity Index (MMI), Rheumatic Disease Comorbidity Index (RDCI). The same comorbidity has a different prevalence rate is due to the ICD code defined by different formulas of index (Appendix A).

**Table 2 jcm-10-05460-t002:** Incidence rates (IRs) per 1000 patient months and the incidence rate ratios (IRRs) for the accumulation of any disease in the comorbidity index before, during, and after the diagnostic period in patients with incident rheumatoid arthritis in the period from 2001 to 2008 in Taiwan.

Comorbidity Indexes	Before the Diagnostic Period		During the Diagnostic Period		After the Diagnostic Period		IRR (95% CI)	
	**No. of** **Events**	**Crude IR**	**No. of** **Events**	**Crude IR**	**No. of** **Events**	**Crude IR**	**During Vs. Before the Diagnostic Period**	**After Vs. Before the Diagnostic Period**
CCI	1311	0.007	2353	0.013	2062	0.012	1.80 (1.68 to 1.92)	1.57 (1.47 to 1.69)
ECI	4099	0.023	11,333	0.064	10,011	0.057	2.77 (2.67 to 2.87)	2.44 (2.36 to 2.53)
MMI	2249	0.013	2985	0.017	2828	0.016	1.33 (1.26 to 1.40)	1.26 (1.19 to 1.33)
RDCI	3601	0.020	5309	0.030	4728	0.027	1.47 (1.41 to 1.54)	1.31 (1.26 to 1.37)

Abbreviations: Charlson Comorbidity Index (CCI), Elixhauser Comorbidity Index (ECI), Multimorbidity Index (MMI), Rheumatic Disease Comorbidity Index (RDCI). During the diagnostic period indicates the period of 4 months before and after the initial diagnosis.

**Table 3 jcm-10-05460-t003:** One-year and 5-year mortality analyses of the four comorbidity indexes.

Comorbidity Indexes	Patient Number (%)	Mortality Rate (Per 1000 People)		Crude HR (95% CI) for Death		Age- and Sex-Adjusted HR (95% CI) for Death	
		1-Year	5-Year	1-Year	5-Year	1-Year	5-Year
**CCI**							
**Low score (0–1)**	**20,244 (81.7)**	**3.2**	**41**	**1.0 (Reference)**	**1.0 (Reference)**	**1.0 (Reference)**	**1.0 (Reference)**
**High score (≥2)**	**4523 (18.2)**	**22.8**	**175.8**	**7.3 (5.3–9.9)**	**4.6 (4.2–5.1)**	**4.3 (3.1–6.0)**	**2.4 (2.1–2.6)**
**ECI**							
**Low score (0–3)**	**16,720 (67.5)**	**3.2**	**38.4**	**1.0 (Reference)**	**1.0 (Reference)**	**1.0 (Reference)**	**1.0 (Reference)**
**High Score (≥3)**	**8047 (32.5)**	**14.2**	**122.3**	**4.5 (3.2–6.2)**	**3.3 (3.0–3.7)**	**2.9 (2.1–4.1)**	**2.1 (1.9–2.3)**
**MMI**							
**Low score (0–1)**	**14,679 (59.3)**	**2.4**	**30**	**1.0 (Reference)**	**1.0 (Reference)**	**1.0 (Reference)**	**1.0 (Reference)**
**High score (≥1)**	**10,088 (40.7)**	**13.1**	**117.1**	**5.5 (3.8–8.0)**	**4.0 (3.6–4.5)**	**3.0 (2.0–4.5)**	**1.9 (1.7–2.2)**
**RDCI**							
**Low score (0–2)**	**16,132 (65.1)**	**3**	**33**	**1.0 (Reference)**	**1.0 (Reference)**	**1.0 (Reference)**	**1.0 (Reference)**
**High score (≥2)**	**8635 (34.9)**	**13.7**	**126.6**	**4.5 (3.2–6.3)**	**4.0 (3.6–4.5)**	**2.5 (1.7–3.5)**	**2.0 (1.8–2.2)**

Abbreviations: Charlson Comorbidity Index (CCI), Confidence interval (CI), Elixhauser Comorbidity Index (ECI), Hazard ratio (HR), Multimorbidity Index (MMI), Rheumatic Disease Comorbidity Index (RDCI).

**Table 4 jcm-10-05460-t004:** The discriminant capacity of the four comorbidity indexes for the 1- and 5-year survival in patients with RA.

Models	1-Year Mortality		5-Year Mortality	
	Harrell’s C-Statistics	AIC	Harrell’s C-Statistics	AIC
Base model	0.744	1868	0.777	10,281
Base model + CCI	0.796	1783	0.802	9879
Base model + ECI	0.772	1829	0.793	10,024
Base model + MMI	0.779	1821	0.792	10,038
Base model + RDCI	0.773	1817	0.791	10,048

Abbreviations: Charlson Comorbidity Index (CCI), Elixhauser Comorbidity Index (ECI), Multimorbidity index (MMI), Rheumatic Disease Comorbidity Index (RDCI). The base model included age, sex, income quartile, urbanization, and occupation groups. The Harrell’s c-statistics indicates the prediction models, which are as follows: 0.5 (as well as chance), 0.7–0.8 (acceptable), 0.8–0.9 (excellent), and 0.9–1 (outstanding prediction). The AIC statistics were calculated, and a small AIC indicates the better predictive ability of the model.

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
