# Peer review of "Comparison of Indexes to Measure Comorbidity Burden and Predict All-Cause Mortality in Rheumatoid Arthritis"

_jcm, 2021, doi:10.3390/jcm10225460_

Round 1
Reviewer 1 Report
I agree with authors Good paper.
Author Response
Thank you for your kindly revision and accomplishment.
Reviewer 2 Report
The time, which is analyzed in Taiwan retrospectivly 1998 - 2008. So there should be some information when TNF Inhibitors were avalilable. Only 0,2% of the RA patients were treated with a therapy, which is state of the art nowadays.
Leflunomide 105 (1.0)
Cyclosporine 165 (1.5)
TNF inhibitor 17 (0.2)
This does not represent RA patients in the last 10 years, much more TNFi, much more LEF, much less CSA
So one major limitation is, that this population you don´t see anymore in daily routine. It would be intersteing, if there is a difference between 1998-2003 and 2004-2008. And you should comment on the very low rate of TNFi biological therapy.
And in the discussion I miss a word about comorbidityin the genral population. If patients get older, they will get more comobidities, that´s the circle of life (circle of getting older). So there are data avalilable about comorbidity which has to be set in correlation to the comorbidity of RA patients. It´s higher we know, but ist should be adressed.
Author Response
Comment 1:
The time, which is analyzed in Taiwan retrospectively 1998 - 2008. So there should be some information when TNF Inhibitors were available. Only 0,2% of the RA patients were treated with a therapy, which is state of the art nowadays. So one major limitation is, that this population you don´t see any more in daily routine. It would be interesting, if there is a difference between 1998-2003 and 2004-2008. And you should comment on the very low rate of TNFi biological therapy. This does not represent RA patients in the last 10 years, much more TNFi, much more LEF, much less CSA.
Response 1:
The medications in table 1 are initial medications used within three months since RA was diagnosed. Hence, the rate of TNF inhibitors, cyclosporine and leflunomide therapy are very low. We already added this description in the part of “Data Source And Patient Cohort” in METHODS.
Comment 2:
And in the discussion I miss a word about comorbidity in the general population. If patients get older, they will get more comorbidities, that´s the circle of life (circle of getting older). So there are data available about comorbidity which has to be set in correlation to the comorbidity of RA patients. It´s higher we know, but it should be addressed.
Response 2:
We added last paragraph in discussion about comorbidity index in control groups: ” The characteristic of aging is that it is easy to get many chronic diseases, which means that there is a higher chance of getting comorbidities. Aging can cause chronic dysregulation and even further dysfunction in multiple organ systems, so the comorbidities can be founded by doctors. In elderly populations, the prevalence of patients with multimorbidities increases.[43] The comorbidity index rise up with time in general population was also demonstrated.[44] RA patients have higher score of four comorbidity indexes compared with control group. The reason may be due to RA patients are in a higher status of inflammation compared to general population.”
This manuscript is a resubmission of an earlier submission. The following is a list of the peer review reports and author responses from that submission.
Round 1
Reviewer 1 Report
Dear authors, you provided data on the rate and impact of comorbidities in patients with RA showing an increased mortality predicted by higher comorbidity scores. In addition, with diagnosis of RA, comorbidities increased.
All data are from a nationwide registry and covered the whole population, all deaths should be seen. In addition only comorbidities documented at least twice are calculated. Limitations are missing comparisons to non RA population and missing data on RA disease activity, treatment and response reducing the significannce of the work.
The limitations are named, thus the data are trustable.
Table 1 shows 2 results for some comorbidities e.g. hypertension without explanation in that line, probably due to different classifications as mentioned below. Thas should be clarified in the line.